# Detecting Patient Position Using Bed-Reaction Forces for Pressure Injury Prevention and Management

**DOI:** 10.3390/s24196483

**Published:** 2024-10-09

**Authors:** Nikola Pupic, Sharon Gabison, Gary Evans, Geoff Fernie, Elham Dolatabadi, Tilak Dutta

**Affiliations:** 1KITE Research Institute, Toronto Rehabilitation Institute—University Health Network, Toronto, ON M5G 2A2, Canada; 2Institute of Biomedical Engineering, University of Toronto, Toronto, ON M5S 3G9, Canada; 3Department of Physical Therapy, University of Toronto, Toronto, ON M5G 1V7, Canada; 4Rehabilitation Sciences Institute, University of Toronto, Toronto, ON M5G 1V7, Canada; 5Vector Institute, Toronto, ON M5G 1M1, Canada

**Keywords:** pressure ulcer, machine learning, neural networks, prevention and control, wound healing, patient positioning, pressure

## Abstract

A key best practice to prevent and treat pressure injuries (PIs) is to ensure at-risk individuals are repositioned regularly. Our team designed a non-contact position detection system that predicts an individual’s position in bed using data from load cells under the bed legs. The system was originally designed to predict the individual’s position as left-side lying, right-side lying, or supine. Our previous work suggested that a higher precision for detecting position (classifying more than three positions) may be needed to determine whether key bony prominences on the pelvis at high risk of PIs have been off-loaded. The objective of this study was to determine the impact of categorizing participant position with higher precision using the system prediction F1 score. Data from 18 participants was collected from four load cells placed under the bed legs and a pelvis-mounted inertial measurement unit while the participants assumed 21 positions. The data was used to train classifiers to predict the participants’ transverse pelvic angle using three different position bin sizes (45°, ~30°, and 15°). A leave-one-participant-out cross validation approach was used to evaluate classifier performance for each bin size. Results indicated that our prediction F1 score dropped as the position category precision was increased.

## 1. Introduction

Pressure injuries (PIs), also known as bed sores or pressure ulcers, are defined as “localized damage to the skin and underlying soft tissue usually over a bony prominence or related to a medical or other device” [1]. PIs are largely preventable, yet they affect one in four Canadians across all healthcare settings [2]. The current practice for preventing PIs is to minimize the risk of prolonged tissue deformation by repositioning patients regularly to allow the compressed tissues to return to their normal state [1,3,4,5,6,7,8]. Unfortunately, evidence suggests that adherence rates to repositioning schedules are poor in clinical environments [6,7,9,10,11,12]. We expect that patients in the home are repositioned even less than in clinical environments due to lack of evidence-based educational materials and tools directed at unpaid home caregivers [13,14,15]. Furthermore, the unpredictability in the home environment may also pose a barrier to adherence to proper protocols and device use compared with a more controlled hospital environment [14]. One study showed that slightly less than one-third of patients were correctly using pressure-redistributing devices in the home environment [14].

The cost of PIs to the healthcare system is enormous, with the estimated cost currently exceeding USD 26.8 billion in the United States [16]. In the UK, approximately USD 4.5–5.1 billion is spent on wound management annually, and two-thirds of this cost is spent on treating wounds in the community setting [17]. Extrapolating from studies conducted in the UK, the average cost for treating PIs not requiring hospitalization is approximately USD 3000 per month in Canada, and the cost increases by an order of magnitude to approximately USD 30,000 per month if hospitalization is required [18,19,20,21,22].

A recent study from the United States found that the prevalence of all at-risk patients presenting to the hospital with a PI was 7.4% [23]. This value is higher than the 2.3% to 5.8% admission rates of two previous studies using electronic databases and greater than the 6.6% prevalence from a study using surveillance methodology [24,25,26]. In another study from the UK, it was found that 81% of patients presenting to the hospital with community-acquired PIs were presenting from a private residence [27].

Therefore, there is a strong need for a study of repositioning in the home environment for the prevention and management of PIs. However, existing methods for tracking repositioning are limited. Leveraging artificial intelligence (AI) could provide an easy solution to gather data and accurately track repositioning. AI has already been used in the clinical healthcare environment to enhance diagnostic accuracy, risk stratification, and treatment efficiency, as well as in the home environment to enhance precision monitoring for personalized medicine [28,29,30,31]. The use of load cells and AI, such as machine learning, may be able to provide a novel non-contact, unobtrusive measurement tool for monitoring patient position.

There are two commercially available technologies that are available for tracking patient positioning in bed in hospitals and nursing homes for PI prevention: pressure mats and body-worn inertial sensors. Pressure mats (e.g., Curiato, XSENSOR) can provide visualizations of the pressure distribution under a patient, which can be used to provide alerts to healthcare providers [6]. Single-use inertial motion sensors (e.g., Leaf Healthcare) attach to a patient’s chest to wirelessly monitor their position and provide an interface to indicate when a patient needs to be repositioned. Each of these technologies has its own limitations. Pressure mats can counteract the benefits of specialized mattresses and have been found to increase the risk of PI by 20% [32]. Pressure mats can also increase the risk of spreading bacteria from infected wounds [33]. Similarly, inertial sensors require attachment to the patient’s skin or clothing using adhesives or some other attachment method, which can be problematic. If adhesives are too strong, they can damage a frail patient’s skin. If the attachment method is not perfectly secure, a sensor can become detached and end up under the patient, potentially causing a PI.

In addition, there is a relatively new type of flexible electronics designed to track positioning during sleep that may be useful for pressure injury prevention [34,35,36]. These devices may have some of the same limitations as pressure mats, but may be beneficial if they can be made to be thin and easy to clean.

### The Use of Loadcells to Detect and Classify Patient Movements

There has been some previous work using load cells and machine learning to classify patient movements in bed. Duvall et al. found that a K-nearest neighbor classification system was able to classify four movements in bed (rolls, turns in place, extremity movements, and caregiver-assisted turns) with 94.2% accuracy [37]. Similarly, Adami et al. classified movements into the three categories based on movements of the torso and lower extremities and achieved a 97% classification accuracy [38]. Alaziz et al. developed a load-cell-based system to detect and classify big and small movements using log-peak values and a threshold-based algorithm that achieved error rates of 6.3% and 4.2%, respectively [39]. This team classified nine classes of movements using a multi-level binary decision tree paired with a Support Vector Machine (SVM) at each decision step to achieve an accuracy of 90% and they applied SVM, random forest, and XGBoost techniques separately to classify the movements and then combined the techniques to achieve a 91.5% accuracy [40,41]. Minteer et al. also designed a system that can detect if a patient movement has occurred with 85% reliability; however, the system did not provide any additional information such as type of movement or what position the patient was in [42]. An important benefit of this study was that the system was tested on a real clinical population that was immobile and needed assistance in repositioning.

Overall, detecting movement in bed may be clinically useful, as there needs to be a detected movement for an individual’s position to change. However, these systems are limited as they have only demonstrated their ability to correctly classify a handful of predetermined movements. They may leave out multitudes of other movements that the system will need to contend with (e.g., dressing, changing incontinence briefs, etc.), which may cause false positive classifications of repositioning events. Furthermore, detecting a movement does not mean that an offloading event has occurred, as an individual could return to the same position and load the same tissues. Capturing an offloading event would ensure that the compressed tissue is allowed to return to its nature state, which decreases the risk for developing a PI.

Reflecting on the above, our team has developed a system that uses machine learning to detect the position of a patient based on data from load cells under each bed leg. We believe our approach has the potential to be more robust than previous movement-based approaches because it categorizes patient position when steady-state values are detected between movements (i.e., when the individual is at rest). This ensures that we capture offloading events because we are able to compare the current position to a previous position and decide if there has been enough of a change to offload the area of interest. Further, it is important to note that some positions are more likely to put patients at a higher risk of PIs than others. For instance, an individual that is lying on their side at 90° to the surface of the bed is likely to increase the forces acting on their greater trochanter compared to a position where the individual is lying on their side at a lower angle (e.g., 30–60° from supine).

Our proof-of-concept work was able to detect healthy participant position (supine, left-side, or right-side) with 94.2% accuracy (*n* = 20) [43]. However, when we tested this model on the data collected from nine older adults sleeping in their own beds at home, we noted there was a large variation in sleeping positions that were adopted at home that were not seen in the laboratory environment [44]. In particular, the larger variability in sleeping positions raised the question of whether it may be necessary to predict an individual’s position in bed with higher precision than only three categories (supine, left, or right). A companion paper will compare the results of this study with the ranges of positions for which key bony areas of the pelvis are off-loaded to determine the optimal position category precision that is needed [45]. Therefore, the objective of this study was to determine the impact of categorizing participant position with higher precision (i.e., more than three position categories) on the system prediction F1 score. We hypothesize that there will be a drop in prediction accuracy when an individual’s position is predicted with a higher precision than the three categories of left-side lying, right-side lying, and supine.

## 2. Materials and Methods

This study was approved by the Research Ethics Boards of University Health Network (Protocol# 17-5140).

### 2.1. Participants

A convenience sample of 20 healthy participants (10 males, 10 females) was recruited for this study. Two participants were excluded from the analysis due to an IMU malfunction, leaving data from a total of eighteen participants for data analysis. All participants provided informed consent.

### 2.2. System Setup

The instrumentation was set up in a similar manner to Wong et al. [43] and Gabison et al. [44]. Data was collected in a simulated patient care environment (CareLab) located at the KITE Research Institute, Toronto Rehabilitation Institute—University Health Network. Single-axis load cells comprised of four load sensors (model DLC902-30KG-HB, Hunan Detail Sensing Technology, Changsha, Hunan, China) arranged in a full Wheatstone bridge circuit were placed under each of the four wheels of a hospital bed (Carroll Hospital Group, Kalamazoo, MI, USA). Figure 1 and Figure 2 are visual representations of the system setup. The load cell signals were amplified, filtered, and converted from analog to digital using a signal conditioner (GEN 5, AMTI, Watertown, MA, USA) configured for 5.0 VDC excitation and a gain of 500 for each channel. NetForce software (version 3.5.2, AMTI, Watertown, MA, USA) running on a laptop PC (Thinkpad T520, Lenovo, Hong Kong, China, 2.5 GHz Intel Core i5 CPU, 4 GB of RAM) was used to collect the load cell data at 50 Hz with 16-bit resolution. An overhead video camera was also positioned above the bed to capture ground truth video of the participant positions.

Participants were fitted with two inertial measurement units (IMU) (Shimmer3, Shimmer Sensing, Dublin, Ireland) at the pelvis. One of these IMUs was used to log the ground truth pelvic angles and the other was used to view the angle in real-time to ensure the participant was positioned appropriately. Two sensors were required because our IMUs did not allow for logging and real-time viewing simultaneously. The IMU was connected to a laptop (Aspire 5, Acer, Xizhi, New Taipei, Taiwan, 1.8 GHz Intel Core i7 CPU, 12 GB of RAM) via Bluetooth to ConsensysPRO (version 1.6.0, Consensys, Dublin, Ireland) to collect data at 256 Hz.

### 2.3. Defining Participant Position While in Bed

The participant’s position in bed was defined by the angle of the pelvis in the transverse plane using an imaginary reference line perpendicular to the line connecting the left and right anterior superior iliac spines in the frontal plane. The angle between this reference line and a line perpendicular to the surface of the bed was defined as the transverse pelvic angle (TPA). This angle was defined such that the TPA would be 0° when participants were supine, a negative value when participants rolled onto their left side, and a positive value when they rolled onto their right side. Figure 3 shows a visual representation of the TPA.

### 2.4. Defining Bin Sizes

A total of three different bin sizes were used to test the effects of increased precision on performance (Figure 4). The 45° bin size is seen in Figure 4a, the 30° bin size is seen in Figure 4b, and the 15° bin size is seen in Figure 4c. For all data, the supine position was defined as any angle between −22.5° and +22.5°.

### 2.5. Data Collection

The data was collected in two phases. The position of participants was monitored using the real-time angle values from one of the IMUs attached to the participant’s pelvis. One of the researchers would gently reposition participants if they deviated more than ±5° from the target angle.

In Phase One, participants were instructed to cycle through a series of 11 unique TPA positions at 0°, ±15°, ±30°, ±45°, ±60°, and ±90°. In Phase Two, participants were instructed to assume any 10 random positions they chose from −180° to +180° to reflect the wide range of positions that can be adopted in bed.

All positions were held for three minutes. Figure 5 shows the order of positions participants were asked to assume, including the intermediate positions (when participants returned to supine) that were held for one minute. The intermediate positions served as a method of standardizing the insertion and removal of pillows for the primary phase.

### 2.6. Data Supplementation

The resulting dataset consisted of 2458 observations. Additional data was incorporated into the training set to increase the size of the data set. This additional data was only used to train machine and deep learning classifiers for the detection of supine, left, or right positions. Supplemental dataset A (*n* = 4910) was from the data collected by Wong et al. in a lab environment [43]. Supplemental dataset B (*n* = 13,913) was from data collected in the home environment by Gabison et al. [44]. The total dataset for Phase One included 21,281 observations.

In our Phase Two dataset, we include only the correct predictions from Phase One (*n* = 2002), and classify them into smaller Phase Two left (*n* = 763), supine (*n* = 622), and right (*n* = 617) datasets. In Phase Two, we focus only on the left and right datasets. Table 1 illustrates which datasets were used for Phase One and Phase Two classifications.

### 2.7. Data Processing

#### 2.7.1. Load Cell Data

Load cell data was processed in the same manner as Wong et al. [43] and Gabison et al. [44]. Load cell data was exported from Netforce and processed using MATLAB 2020a (MathWorks, Natick, MA, USA). The data was manually sorted into trials by removing segments of data where positional changes occurred. The center of mass (CoM) of the bed and patient was calculated using Equations (1) and (2), where *CoM_x* and *CoM_y* reflect the *CoM* parallel to the width and length of the bed, respectively. In Equations (1) and (2), *LH* and *RH* represent the vertical forces experienced by the left and right sensors at the head of the bed, respectively; *LF* and *RF* represent the vertical forces by the left and right sensors at the foot of the bed, respectively; and *l* and *w* refer to the distance between the load cells along the length and width of the bed, respectively.
(1)$$CoM\_x=\frac{w}{2}\times \frac{LH+LF-RH-RF}{LH+LF+RH+RF}$$
(2)$$CoM\_y=\frac{l}{2}\times \frac{LH+RH-LF-RF}{LH+LF+RH+RF}$$

The *CoM_x* and *CoM_y* signals were low-pass-filtered with personalized Chebyshev Type II filters in order to isolate the changes in CoM signals related to respiration. The MATLAB filtfilt function was applied, which ensures a zero-phase shift, to obtain *CoM_resp_x* and *CoM_resp_y*. The maxima (*tmax*) and minima (*tmin*) in the *CoM_resp_x* and *CoM_resp_y* signals were identified by finding the zero crossings for the first derivative of each signal, where the maxima and minima coincide with the end of each exhalation and inhalation, respectively [46]. Equation (3) was used to calculate the angle of the principal axis of the ellipsoid traced by the resultant *CoM_resp* signal relative to the positive x axis for each tmax and subsequent *tmin*.
(3)$$CoM\_resp\_ANG=arctan\left|\frac{{CoM\_resp\_y}_{\left(tmax\right)}-{CoM\_resp\_y}_{\left(tmin\right)}}{{CoM\_resp\_x}_{\left(tmax\right)}-{CoM\_resp\_x}_{\left(tmin\right)}}\right|$$

Every data point used for the training and testing of the machine learning and deep learning classifiers was obtained by taking the average of a 45 s window, containing 2250 observations. New values were computed by shifting the window by 15 s. Approximately 10 data points were calculated from each pose for each participant. Missing data from one participant was interpolated based on values from the same window.

#### 2.7.2. IMU Data

The ground truth data from the IMUs were classified into one of three positions (right-side lying, left-side lying, or supine) based on their Euler angle values. The IMU data were further classified four more times using the generated Euler angles, once for each of the different bin sizes, to allow for more precise TPA detection.

#### 2.7.3. Supplemental Data

The supplemental data were previously classified into one of the three positions (right-side lying, left-side lying, or supine) by reviewing video data collected from supplemental dataset A and by three independent and blinded reviewers using time-lapsed images from supplemental dataset B.

### 2.8. Feature Selection

Table 2 shows the features extracted from load cell data that paralleled those used by Wong et al. [43] and Gabison et al. [44]. Feature importance was reassessed, showing that *ratio_rmsCoM_resp*, *CoM_resp_ANG*, and *stdCoM_y* were the three most important features. This finding is in line with our previous work that also demonstrated that the most important part of the signal from our load cells had to do with the small changes in the centre of mass of the participant that occurred with each breath. See Appendix A for a graphical representation that shows the relative importance of all 12 features.

### 2.9. Two-Phase Machine Learning Approach

In order to predict positions more precisely, a two-phase hierarchical classification approach was used, similar to the one used by Liang et al. [47]. In Phase One, we classified participant’s position as left-lying, supine, or right-lying, as previously defined in Section 2.3. Phase One was trained on the total data set and tested only on the PI dataset. Any incorrect classifications from Phase One were marked as incorrect and removed from the dataset. The remaining output of correct classifications from Phase One was divided into smaller datasets for left, supine, and right, as explained in Section 2.4, with the left and right datasets serving as the input for Phase Two.

We chose to remove the incorrect classifications from Phase Two as this is a pilot study that had limited data, so we prioritized the ability to predict position with more precision. Furthermore, as we defined the supine dataset to be a constant range, it could not be analyzed with more precision and was thus not used in Phase Two.

In Phase Two, the algorithm was used to predict the participant’s position more precisely. The side-lying positions were classified with increasing precision by dividing the range between ±22.5° and ±112.5° into two, three, and six bins, which correspond to bin sizes of 45°, 30°, and 15°, respectively. This can be seen in Figure 4a–c. The classification was repeated three times, once for each bin size. The range of positions was limited to positions between ±112.5° because there was not enough data to reliably predict positions outside of this range.

By splitting the classification into two phases, it allowed additional data to be used as Phase One classification did not require associated IMU data. In this way, the model was able to be trained on more data to narrow down a portion of its classification task.

Please see Appendix A for a detailed example of how Phase One and Phase Two classification are integrated.

#### 2.9.1. Leave-One-Participant-Out

A leave-one-participant-out cross validation approach was used to evaluate the accuracy of the classifier while maximizing the number of training observations. Using this method, a classifier was trained on the new data set that incorporated 17 participants and tested on the 1 excluded participant. This procedure was repeated 18 times, once for each participant. The overall performance measures were estimated from the averaged errors for each individual test sample.

#### 2.9.2. Incremental Learning

Incremental learning was used to evaluate the potential of the classifier to adapt to the left-out participant. The classifier was trained using different percentages of the left-out participant’s data (c = 0%, 10%, 20%, 30%). To maintain a uniform test set, the left-out participant’s data was split into a 70% test set and a 30% incremental learning set, from which different c values were added to the training set. The best performing incremental learning level (ILL) was used to create the smaller left and right datasets for the more precise step two classification.

#### 2.9.3. Machine Learning Classifiers

Table 3 shows a list of models used in both phase one and two classifications. Appendix B contains tables that describe the architectures of all the created models.

### 2.10. Statistical Analysis

A Shapiro–Wilk test and a Levene test were used in all cases to assess the normality and the homogeneity of the data, respectively. Both were not satisfied, resulting in non-parametric tests being used.

#### 2.10.1. Phase One Classification

A Friedman’s ANOVA was used to identify any significant differences in the mean F1 score and accuracy based on the best performing ILL. Post hoc Wilcoxon Rank Sum tests with Bonferroni corrections for three comparisons (*p* < 0.0167) were used to compare the top three performing models.

A Friedman’s ANOVA was also used to compare the ILLs for each of the top models to determine their impact on performance. Post hoc Wilcoxon Rank Sum tests with Bonferroni corrections for three comparisons (significance needed *p* < 0.0167) were used to compare each incremental learning level to its adjacent value(s) (i.e., 0% to 10%, 10% to 20%, and 20% to 30%).

#### 2.10.2. Phase Two Classification

ANOVAs were used to compare the classification results to determine the effect of bin size on model performance for left and right classification. The best classifier from each bin size was compared to its adjacent bin size(s) (i.e., 45° to 30°, and 30° to 15°). Post hoc Wilcoxon Rank Sum tests with Bonferroni corrections for two comparisons (*p* < 0.025), were used to compare the top three performing models.

## 3. Results

### 3.1. Participants

Descriptive statistics of all the participants recruited for this study are shown in Table 4.

### 3.2. Phase One Classification

The best-performing model was the LGB classifier with a F1 score of 0.8343 ± 0.1362 at an ILL of 30%. All other model scores and associated graphs are found in Appendix C.

#### 3.2.1. Comparing Machine Learning Models

The data was confirmed to be non-parametric with a significant difference between the mean F1 score, χ2(8) = 96.398, *p* < 2.2 × 10^−16^. The top three performing classifiers were LGB, GBC, and MLP 2. The post hoc comparisons with the LGB model were found to be statistically significant: LGB vs. MLP 2: V = 27, *p* = 0.0115; LGB vs. GBC: V = 156, *p* = 0.0023; GBC vs. MLP 2: V = 69, *p* = 0.740.

#### 3.2.2. Comparing Incremental Learning Levels

As the LGB model was shown to be the best model, the ILLs were compared. The LGB data was confirmed to be non-parametric with a significant difference between the mean F1 score, χ2(3) = 28.787, *p* < 2.483 × 10^−6^. Appendix C contains a graphical comparison between the mean F1 score of the different ILLs for the LGB model. Two out of three post hoc comparisons were found to be statistically significant, 0% vs. 10%: V = 10, *p* = 0.00109; 10% vs. 20%: V = 40, *p* = 0.155; 20% vs. 30%: V = 13, *p* = 0.00172. Therefore, the LGB classifier at an ILL of 30% was the best-performing model. It was used to create the left and right data sets for Phase Two.

### 3.3. Phase Two Classification

Table 5 shows the overall mean F1 score, accuracy, and standard deviation values for the best model, LGBoost, across all 18 participants from Phase Two left and right classification using an ILL of 30%.

#### 3.3.1. Comparing Left Bin Sizes

The data for Phase Two left classification was confirmed to be non-parametric with significant differences between the F1 scores of different bin sizes, χ2(1) = 54, *p* < 2.00 × 10^−13^. Figure 6 shows a visual comparison between the different bin sizes for left-sided lying.

Post hoc comparisons were found to be statistically significant for 45° vs. 30° bin size, V = 147, *p* = 0.00789, but not for 30° vs. 15° bin size, V = 108, *p* = 0.142.

#### 3.3.2. Comparing Right Bin Sizes

The data for Phase Two right classification was confirmed to be non-parametric with significant differences between the F1 scores of different bin sizes, χ2(1) = 54, *p* < 2.00 × 10^−13^. Figure 7 shows a visual comparison between the bin sizes.

Post hoc comparisons for mean F1 was found to be statistically significant for 45° vs. 30° bin size, V = 118, *p* = 0.0105, but not for 30° vs. 15° bin size, V = 111, *p* = 0.0280.

## 4. Discussion

This study aimed to improve our non-contact position detection system for preventing PIs by increasing the precision of position classification. By analyzing data from load cells and IMUs, we assessed how classifying individual positions with finer granularity (45°, ~30°, and 15°) impacts the system’s predictive accuracy. The study found that higher precision in detecting positions led to a decrease in the F1 score of the system. This highlights a critical trade-off between detailed position tracking and predictive reliability, contributing valuable insights into optimizing PI prevention technologies.

### 4.1. Phase One Classification

The best performing model was the LGBoost. This F1 score is an improvement over the F1 score of 0.6841 that Wong et al.’s previous model was able to achieve on this data [43]. It is likely that the F1 score represents an underestimate of the true F1 score of our new system as the training set includes data without the IMU ground truth data for Phase One. Additionally, we combined data from different settings, which may have negatively impacted the performance of our model as there was more variability to be accounted for by the model, such as the type of bed, size of bed, and carpeted floors. This likely explains why Wong et al.’s model was able to perform well on their lab-controlled dataset but had its performance drop significantly in this mixed dataset.

The Leaf Healthcare system, which attaches an IMU to the patient’s body, will consistently record a real-time angle for patient position. However, this comes at the cost of having an adhesive placed on the skin that could cause problems itself. Further, Leaf Healthcare’s system is advertised as placing the sensor on a patient’s chest, which may not accurately represent position changes and offloading as the hips can rotate independently of the chest. By focusing on predicting the TPA, we believe that our system would more accurately reflect the true position needed to be monitored for offloading.

#### Comparing Incremental Learning Levels

Our results suggest that collecting data for incremental learning has the potential to better personalize the model to individuals at high risk of developing PIs, with the potential to improve their care. It is important to note that the test set for each participant in the study was limited, ranging from 81 to 206 observations (mean 136.6 observations), meaning that a maximum of 61 observations at 30% incremental learning were added to a data pool of ~20,000 observations for incremental learning. Considering that the incremental learning, with so few observations, was able to improve the overall F1 score by slightly over 0.07, for the LGB model, it would be important to further investigate the impacts of incremental learning with a larger data set. Additionally, it would be important to investigate whether the statistical significance of incremental learning translates to clinical significance and an improved prevention of PIs.

### 4.2. Phase Two Classification

The LGB model performed significantly better than other models for all test cases in Phase Two. As we removed the incorrect classifications from Phase One before feeding the data into Phase Two, it is important to interpret the F1 score and accuracy as an overestimation. To get a hypothetical maximum F1 score, we would need to multiply the F1 score from Phase One by the F1 scores from Phase Two. This results in a hypothetical maximum F1 score of 0.767 for left using 45° bins and our hypothetical minimum of 0.617 for right using 15° bins. It is difficult to compare Phase Two to other position-based studies as they focused only on predicting left, right, or supine positions, whereas our study increases the precision. As hypothesized, this inherently decreased our F1 score and accuracy compared to other studies. We also identified a trend of left-side positions being predicted correctly slightly more often than right-side positions. The reason for this finding is currently unclear.

#### Comparing Bin Sizes

We noted that with increasing number of bin sizes, the accuracy of prediction decreased. However, this was only significant in the Level 2 left and right mean F1 score between 45° vs. 30° bin sizes. It was also significant for left mean accuracy between 45° vs. 30° and for right mean accuracy between 30° vs. 15° bin sizes. These results are important because they indicate that there is likely a trade-off between accuracy and precision when making predictions on estimating an individual’s position in bed. It will be important to optimize the bin size for this system to ensure it is recording and classifying movements of interest. The bin size should be optimized based on information gathered from offloading data and clinical expertise to decide what is the smallest positional change that needs to be captured to offload areas on the pelvis that are at high risk of developing PIs. Our companion paper uses offloading data to draw conclusions about the optimal bin size [45].

### 4.3. Study Limitations and Future Work

This study included some technical limitations. The load cell data was filtered using customized parameters for each participant, which may have led to an increased performance compared to generic parameters. Most training data did not contain IMU ground truth, which may have negatively impacted accuracy. We found that our sample size of 18 and the number of data points (~2500) was too small to perform meaningful one-shot learning or regression analysis. Additionally, the small sample size resulted in removing incorrect predictions from Phase One before feeding the data into Phase Two as we did not have enough data to learn from the incorrect predictions. There was also an imbalance of positions recorded, resulting in fewer positions that were >90° and <−90° for Phase Two classification, which could have negatively impacted performance for predicting positions within that range. Lastly, the hyperparameter tuned architecture was overly complicated, with too many layers for our dataset, and may have overfit the model.

Most of the above limitations can be addressed by collecting additional data. As such, our future work will include collecting longer-duration overnight data with participants in their own homes. After collecting additional data, we plan to treat the detection of patient position as a regression task instead of a classification task and compare their respective performances to identify which approach is better. A regression analysis would strengthen our ability to predict the actual angle rather than an angle bin, which may be exaggerating the errors from our system. Moreover, with additional data, we would be able to learn from incorrect predictions to improve the algorithm’s performance, allowing us to determine the true maximum F1 score attainable. Finally, we plan on further developing this system to allow for real-time prompting so that the device can be implemented in the home setting to monitor offloading.

## 5. Conclusions

The results of this study showed that when trying to predict position more precisely than left side-lying, right side-lying, or supine, the prediction mean F1 score decreased from 0.919 and 0.900 for the left and right ~45° bin size (91.5% and 89.4% accuracy), respectively, to 0.786 and 0.740 for the 15° bins (81.5% and 75.4% accuracy), respectively.

These results demonstrate that our system can track and predict an individual’s positioning in bed and needs to be further tested in a home care setting to evaluate its ability in preventing PIs. This paper should be interpreted along with its companion paper that expands on these results from a wound management perspective [45].

## Figures and Tables

**Figure 1 sensors-24-06483-f001:**
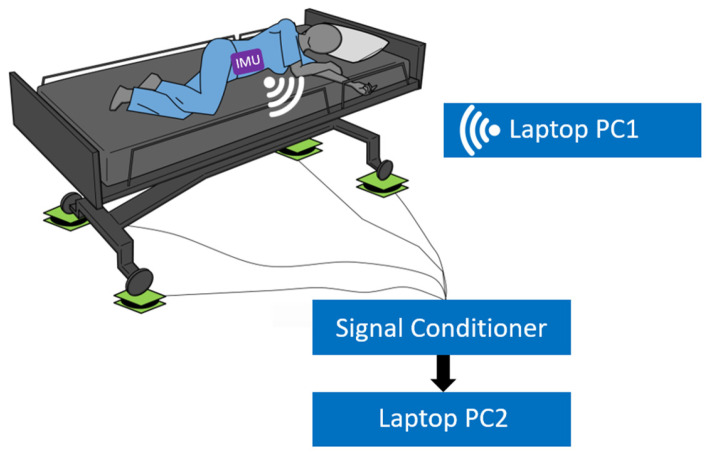
The data collection setup. Four load cells were placed under the legs of the bed. Signals from the load cell were amplified, filtered, and converted from analog to digital using a signal conditioner. Data was saved on a laptop (PC2). IMU data was collected to PC1 via Bluetooth.

**Figure 2 sensors-24-06483-f002:**
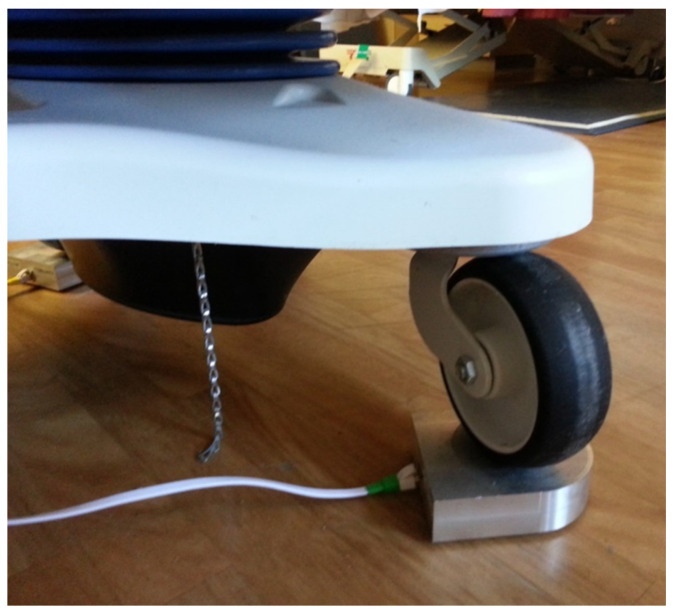
Close-up image of the bed wheel on a load cell.

**Figure 3 sensors-24-06483-f003:**
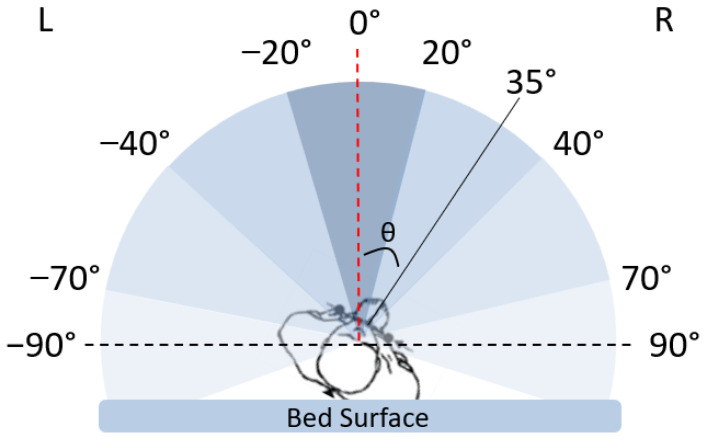
The solid-colored black line represents an individual’s position on the bed in the transverse plane. This line is perpendicular to the line connecting the left and right anterior superior iliac spines on the individual in the frontal plane. The dotted red line represents our true supine reference point at 0° and is defined as the line perpendicular to the surface of the bed. Therefore, the TPA, represented by θ, is defined as the angle between the 0° reference line and the line perpendicular to the frontal plane of the pelvis.

**Figure 4 sensors-24-06483-f004:**

Three different bin sizes were used to test precision on performance: 45° bins (**a**); 30° bins (**b**); 15° bins (**c**).

**Figure 5 sensors-24-06483-f005:**
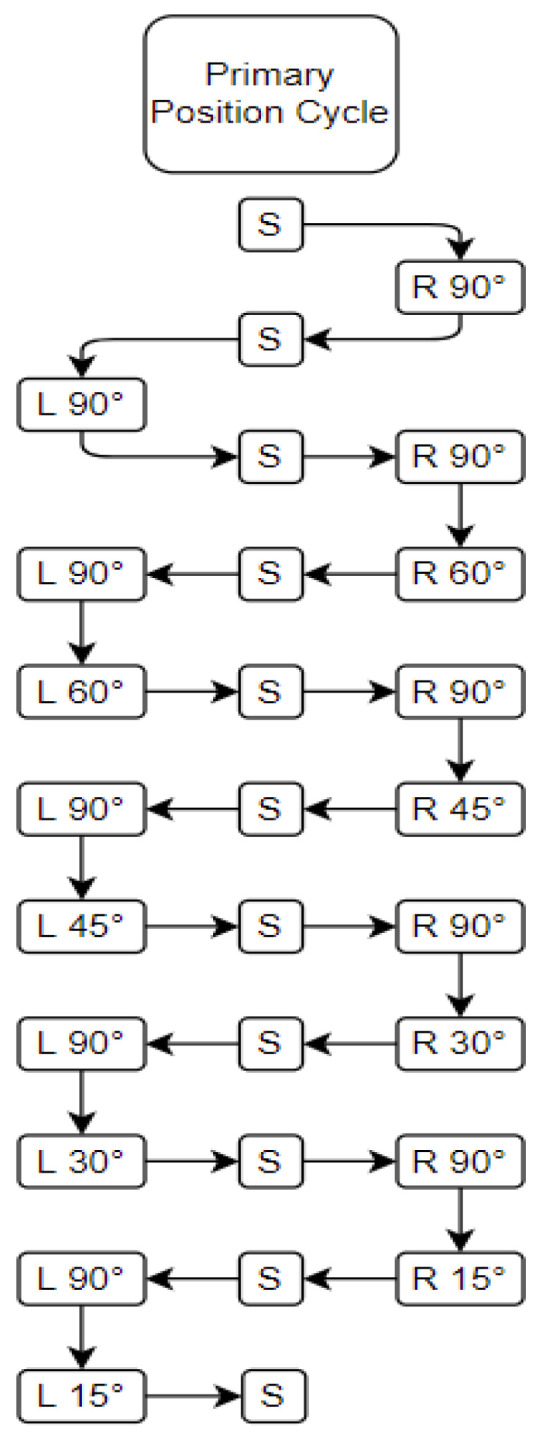
The 11 different positions adopted by patients during primary testing and the order of positions, including intermediate holds at supine (S), right-side-lying at 90° (R90°), and left-side-lying at 90° (L90°).

**Figure 6 sensors-24-06483-f006:**
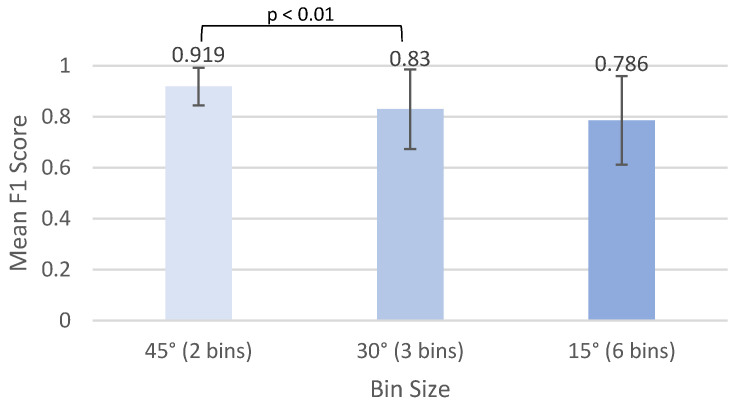
Mean F1 scores for the top models for Phase Two left classification for the three different bin sizes.

**Figure 7 sensors-24-06483-f007:**
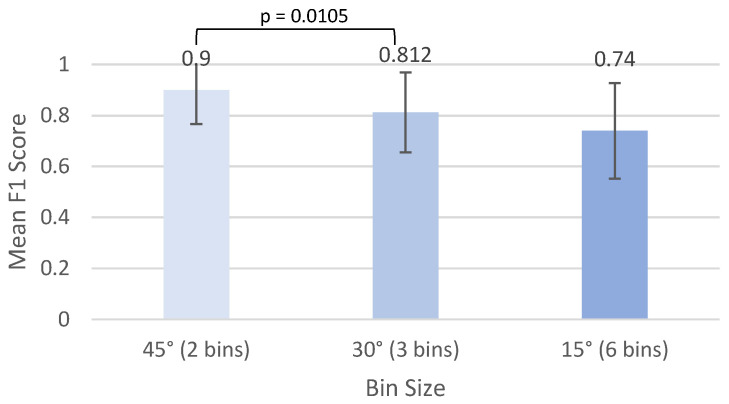
Mean F1 scores for the top models for Phase Two right classification for the three different bin sizes.

**Table 1 sensors-24-06483-t001:** Table describing the different data sources and how they were used.

Data Source	Data Specifications	Data Set Size (Samples)	IMU	Data Use	Phase
Supplemental Dataset A	20 healthy participants	4910	No	Training	One
Supplemental Dataset B	9 healthy participants collected at home and 1 healthy participant collected in a sleep lab	13,913	No	Training	One
PI Dataset	18 healthy participants	2458	Yes	Training and Testing	One
Phase Two Left	18 healthy participants	763	Yes	Training and Testing	Two
Phase Two Right	18 healthy participants	617	Yes	Training and Testing	Two
Phase Two Supine	18 healthy participants	622	Yes	N/A	N/A

**Table 2 sensors-24-06483-t002:** Features extracted by Wong et al. [43] and Gabison et al. [44].

Feature	Description
meanCoM_x	The mean of CoM_x
meanCoM_y	The mean of CoM_y
ratio_meanCoM	The quotient of meanCoM_y divided by meanCoM_x
stdCoM_x	The standard deviation of CoM_x
stdCoM_y	The standard deviation of CoM_y
ratio_stdCoM	The quotient of stdCoM_y divided by stdCoM_x
CoM_resp_ANG	COM angle during inhalation phase only, averaged for all occurrences
stdCoM_resp_ANG	Standard deviation of CoM_resp_ANG
rmsCoM_resp_x	The root mean square of the x-component of CoM_resp during both inhale and exhale phases, normalized to the 97th percentile
rmsCoM_resp_y	The root mean square of the y-component of CoM_resp during both inhale and exhalation phases, normalized to the 97th percentile
ratio_rmsCoM_resp	The quotient of rmsCoM_resp_y divided by rmsCoM_resp_x
rmsPulse	The root mean square of the load cell signals filtered to capture changes resulting from the cardiac cycle

**Table 3 sensors-24-06483-t003:** Table describing the models created, which level they were used for, and the features they used.

Model	Phase One Prediction	Phase Two Prediction	Features Used
Logistic Regression		X	12 features from Wong et al.
Support Vector Machine		X	12 features from Wong et al.
Gradient Boosting Classifier	X	X	Some of the 12 features from Wong et al.
AdaBoost Classifier	X	X	12 features from Wong et al.
XGBoost Classifier	X	X	12 features from Wong et al.
Light Gradient Boosting Machine Classifier	X	X	12 features from Wong et al.
Multilayer Perceptron (x3)	X		12 features from Wong et al.
Recurrent Neural Network: Long Short-Term Memory	X		12 features from Wong et al. and Automatic feature selection
Convolutional Neural Network: 1-Dimensional	X		12 features from Wong et al. and Automatic feature selection

**Table 4 sensors-24-06483-t004:** Table showing the participant demographics. Age ranges are included to protect privacy, but the mean and standard deviation were computed with exact values.

Participant	Sex (M/F)	Age Range (Years)	Height (cm)	Weight (kg)	BMI
1	F	26–30	170.0	92.0	31.8
2	M	71–75	172.7	75.2	25.1
3	M	31–35	181.0	88.7	27.1
4	M	41–45	181.0	99.8	30.4
5	F	51–55	157.5	87.8	35.4
6	F	21–25	157.5	50.3	20.3
7	M	21–25	175.0	73.8	24.1
8	M	26–30	165.0	65.5	24.1
10	M	31–35	198.0	120.3	30.7
11	F	21–25	161.3	62.6	24.1
12	F	21–25	165.1	49.2	18.0
13	F	21–25	175.0	57.4	18.8
14	F	21–25	175.0	60.8	19.8
15	F	21–25	170.0	63.7	22.1
17	M	21–25	188.0	107.6	30.4
18	M	21–25	179.0	81.8	25.5
19	M	16–20	185.4	78.0	22.7
20	M	21–25	183.0	88.5	26.4
Mean ± (SD)		29.8 ± 13.0	174.4 ± 10.6	77.9 ± 19.2	25.4 ± 4.7

**Table 5 sensors-24-06483-t005:** Mean F1 scores and accuracy with standard deviation for the Phase Two left and right classifications of different bin sizes for the best performing model, LGBoost.

Left			
Model	Bin = 45°	Bin = 30°	Bin = 15°
F1	0.919 ± 0.0736	0.830 ± 0.156	0.786 ± 0.174
Accuracy	0.915 ± 0.0802	0.847 ± 0.145	0.815 ± 0.153
Right			
Model	Bin = 45°	Bin = 30°	Bin = 15°
F1	0.900 ± 0.133	0.812 ± 0.157	0.740 ± 0.188
Accuracy	0.894 ± 0.122	0.823 ± 0.142	0.754 ± 0.177

## Data Availability

Data is unavailable due to the restrictions of our research ethics approval.

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
