# Peer review of "Detecting Patient Position Using Bed-Reaction Forces for Pressure Injury Prevention and Management"

_sensors, 2024, doi:10.3390/s24196483_

Round 1

Reviewer 1 Report

Comments and Suggestions for Authors

This manuscript needs reconsideration after a major revision. The authors should resolve following questions to further improve the manuscript.

1. Figure 2 legend was not matched with figure contents. Please modify it.

2. Figure 1 was not clear for reader’s understanding. Could the author describe more about the experimental platform? Illustrating the bed, and subjects’ position, and related definitions using drawing and real optical images.

3. How about the classification accuracy of position?

4. The authors should compare their results with existing studies. This may highlight the advantages of this study.

5. Could the author provide the real-time data of loadcell, when the participants’ position changes?

6. There are many works about the sleeping posture monitoring, using flexible electronics. Could the author discuss more about it?

7. The application of artificial intelligence (AI) in the intelligent healthcare is an emerging direction and field. For instance, intelligent diagnostics of CVD events [Cell Reports Physical Science 4, 101690; npj Digital Medicine (2019) 39]. Could the authors discuss these in the introduction section to highlight the AI application in intelligent healthcare?

Author Response

Please see attachment. Contains comments for both reviewers. Thank you!

Reviewer 2 Report

Comments and Suggestions for Authors

Interesting paper. A few minor grammatical errors and some things that need to be explained better are detailed below.

1. Line 35, I believe you are missing the word 'are' as in it should read: "PIs are largely preventable..."

2. Line 36, I think the word treating should be replaced with preventing.

3. Line 60, Grammatical error. I think is should be: "there is a strong need for A study of...

4. Lines 98-102, I think a discussion on the value of positions vs. movements could be valuable as both are important. Movements are important because being in any position for too long is a risk. However, positions are important because some positions are more risky than others (i.e., supine and 90 degrees put more weight directly on bony prominences compared with positions between 30 and 60 degrees).

5. Lines 163-166, It appears this is described backwards. Logic and the figure captions say (a) is 45 degree bins and (c) is 15 degree bins.

6. 2.5 Data Collection (Beginning on Line 170), How was it determined that the participants were at the correct angle? I would assume the IMU was used, but it wasn't stated. Also, was there a range that was acceptable? For example, if they were instructed to be at 15 degrees, was anywhere between 10-20 degrees acceptable? 

7. Line 269, Prior to this point (line 127) you stated there were 20 participants, but now you say 18. Later (line 310) you state that two were excluded. To avoid confusion, I would state around line 127 that two were excluded from analysis so the 18 on line 269 makes sense.

8. Line 338, The statement in the table caption doesn't make sense. "The bolded values indicated the best performing model for the respective bin size." You only show one model and nothing is bold.

9. 4.1 Phase One Classification (Beginning on line 367). This whole paragraph seems wrong to me unless I am missing something. First, Phase 1 was just left/right/supine and you had ground truth from the videos/images, so the IMU isn't necessary. Second, you state on lines 244-248 that incorrectly classified data from Phase 1 was removed from the Phase 2 analysis, but now you appear to say it was included in the F1 score. If the incorrect data was indeed removed, then these F1 scores for Phase 2 would be overestimations because you removed data that was known to be bad which is unlikely to happen in the real world. 

Comments which could be included in future work/discussion.

Line 238, There are 12 variables extracted from the load cells. Was there any analysis on the relative importance of those variables to the classifications? For example, I would guess that COM_x is far more important than COM_y. 

4.2.1 Comparing Bin Sizes (beginning on line 393), It would also be interesting to note whether close values were more often classified incorrectly or how far off the predictions were. For example, for the 15-degree bins, was a position of say 37 degrees more likely to be put in the 37.5-52.5 bin rather than a position of 25 degrees? Maybe an analysis of how far off the classification was could be done. Either by bins (missed by one bin vs 2) or by degrees (37 degrees in the 37.5-52.5 bin was off by 0.5 degrees but a 25-degree reading in the 37.5-52.5 bin is off by 12.5 degrees). It might be worth looking at a regression analysis to predict the actual angle rather than bins and seeing how far off the angle predictions are. If the average difference between actual degree of the position and the predicted angle is less than say 10 degrees, I think that would be a huge success even if the placement in bins may be incorrect. 

Author Response

(The authors gave the same response as above.)

Round 2

Reviewer 1 Report

Comments and Suggestions for Authors

The authors have addressed my concerns. The current version can be published.